# Virulence Characterization of *Puccinia striiformis* f. sp. *tritici* in China in 2020 Using Wheat *Yr* Single-Gene Lines

**DOI:** 10.3390/jof11060447

**Published:** 2025-06-12

**Authors:** Jie Huang, Xingzong Zhang, Wenjing Tan, Yi Wu, Hai Xu, Shuwaner Wang, Sajid Mehmood, Xinli Zhou, Suizhuang Yang, Meinan Wang, Xianming Chen, Wanquan Chen, Taiguo Liu, Xin Li, Chongjing Xia

**Affiliations:** 1School of Life Science and Engineering, Southwest University of Science and Technology, Mianyang 621010, China; 18681697196@163.com (J.H.); zxzcurtain@163.com (X.Z.); wenjingtan2022@126.com (W.T.); 17828285566@swust.edu.cn (Y.W.); eli6951@sina.com (X.Z.); yangszh@126.com (S.Y.); 2Anzhou District Bureau of Agriculture and Rural Affairs, Mianyang Bureau of Agriculture and Rural Affairs, Mianyang 622650, China; 3State Key Laboratory for Biology of Plant Disease and Insect Pests, Institute of Plant Protection, Chinese Academy of Agricultural Science, Beijing 100193, China; wqchen@ippcaas.cn (W.C.); liutaiguo@caas.cn (T.L.); 4Department of Plant Pathology, Faculty of Agriculture, Pir Mehr Ali Shah Arid Agriculture University, Rawalpindi 46000, Pakistan; sajidm1529@uaar.edu.pk; 5Department of Plant Pathology, Washington State University, Pullman, WA 99164, USA; meinan_wang@wsu.edu (M.W.); xianming.chen@usda.gov (X.C.); 6Agricultural Research Service, Wheat Health, Genetics, and Quality Research Unit, U.S. Department of Agriculture, Pullman, WA 99164, USA

**Keywords:** *Triticum aestivum*, stripe rust, pathogen, race identification, resistance genes, virulence diversity

## Abstract

Wheat stripe (yellow) rust, caused by the fungus *Puccinia striiformis* f. sp. *tritici* (*Pst*), is one of the most threatening wheat diseases worldwide. Monitoring the virulence of *Pst* population is essential for managing wheat stripe rust. In this study, 18 wheat *Yr* single-gene lines were used to identify the virulence patterns of 67 isolates collected from 13 provinces in China in 2020, from which 33 *Pst* races were identified. The frequency of virulence to different *Yr* genes varied from 1.49% to 97.01%, with 4.48% to *Yr1*, 26.87% to *Yr6*, 11.94% to *Yr7*, 95.52% to *Yr8*, 19.40% to *Yr9*, 11.94% to *Yr17*, 2.99% to *Yr24*, 35.82% to *Yr27*, 38.81% to *Yr43*, 97.01% to *Yr44*, 8.96% to *YrSP*, 1.49% to *Yr85*, 95.52% to *YrExp2*, and 7.46% to *Yr76*. None of the isolates were virulent to *Yr5*, *Yr10*, *Yr15*, and *Yr32*. Among the 33 races, PstCN-062 (with virulence to *Yr8*, *Yr44*, and *YrExp2*) and PstCN-001 (with virulence to *Yr8*, *Yr43*, *Yr44*, and *YrExp2*) were the prevalent races, with frequencies of 28.36% and 11.94%, respectively. These results provide valuable information for breeding resistant wheat cultivars for controlling stripe rust.

## 1. Introduction

Wheat (yellow) stripe rust, caused by *Puccinia striiformis* f. sp. *tritici* (*Pst*), is a significant global disease impacting wheat production. This disease is particularly destructive in regions with climates conducive to the disease and where susceptible cultivars are extensively grown [1]. China represents the most prominent and relatively independent epidemic area for wheat stripe rust in the world [2,3,4]. Since the 1950s, China has experienced 15 major stripe rust epidemics, with damaging outbreaks in 1950, 1964, 1983, 1985, 1990, 2002, 2017, and 2020. These outbreaks have collectively resulted in losses of approximately 14 billion kilograms of wheat [1,2,5], necessitating the major replacement of wheat cultivars seven times [6]. The virulence of *Pst* characterized by dynamic changes, and the emergence of new races that overcome existing resistance in wheat cultivars was a primary factor driving these epidemics [7,8].

The resistance of newly developed wheat cultivars to stripe rust often diminishes, necessitating periodic replacement with newer cultivars [9,10]. This cycle is driven by the continuous evolution of virulence in *Pst* races. *Pst* has a complicated lifecycle that encompasses sexual and asexual reproduction in China, which enables it to quickly acclimatize to rehabilitated ecological niches and effectively overcome resistance genes deployed in wheat cultivars [11,12]. Consequently, it is of the utmost importance to implement effective management and control strategies to combat this disease and guarantee the production of premium-quality wheat [1]. In China, the virulence dynamics of *Pst* have been systematically monitored since 1958, when the first race, CYR1, was identified. CYR1 primarily attacks wheat cultivar ‘Bima 1’, which was released in 1951 and widely grown in the 1950s and the early 1960s throughout the country. Within seven years, the resistance of ‘Bima 1’ transitioned from full immunity to susceptibility [13]. Following this, the wheat variety ‘Abo’ and its derivatives, planted extensively in western China, maintained effective resistance for nine years. However, the emergence of races CYR17 and CYR18 in 1974 overcame its resistance, leading to a significant epidemic in 1975 [14]. More recently, ‘Guinong 22’, released in 2004, showed resistance to all known *Pst* races in China until 2009, when race CYR34 appeared, rendering ‘Guinong 22’ susceptible [15]. Races CYR32 and CYR34 have been predominate, responsible for substantial wheat yield losses over the past decade [15,16,17]. These examples underscore the necessity of ongoing monitoring the virulence of *Pst* populations and breeding efforts to manage the continuous threat posed by the stripe rust pathogen. To this direction, race identification remains the most direct method for assessing the virulence of the stripe rust pathogens. Virulence characterization of *Pst* using wheat *Yr* (yellow resistance) single-gene lines is essential for understanding the dynamics of wheat resistance to this pathogen, particularly the evolution nature of the pathogen’s virulence. Near-isogenic lines (NILs) with an ‘Avocet S’ background have demonstrated excellent discriminative ability and are widely used in global *Pst* studies and stripe rust resistant wheat development [18,19].

A set of 18 wheat *Yr* single-gene lines was first established as differentials for identifying *Pst* races in the United States in 2014 [20]. Recently, Chinese researchers have begun using the *Yr* single-gene lines for studying *Pst* virulence. For instance, Li et al. characterized the virulence in the *Pst* population in the Yunnan province and identified 136 isolates as 64 races using the 18 *Yr* single-gene differentials [21]. The use of the *Yr* single-gene lines can provide insights into the virulence composition of *Pst* races related to specific *Yr* genes and thus, enable directly targeting effective resistance breeding and management strategies. More recently, Zhang et al. analyzed *Pst* isolates collected in 2023 in China using this set of *Yr* single-gene differentials and found virulence frequencies ranging from 0 (to *Yr5* and *Yr15*) to 95.1% (to *Yr8*), indicating differences in effectiveness of these genes [22]. These studies also showcased higher differentiation capabilities of the *Yr* single-gene differentials than the cultivar differentials.

In the present study, we aimed to characterize the virulence of *Pst* isolates collected from the wheat growing regions across China in 2020 using the set of 18 *Yr* single-gene lines. Our objectives were to detect emerging virulent races of *Pst* that overcome the resistance genes, their virulence structure, the frequency of virulence genes, and analysis of their geographical distribution. The virulence of *Pst* isolates in this study was compared with that from 2023 which has been published previously [22]. As this project is a part of systematically monitoring *Pst* virulence dynamics in China, the dynamics of *Pst* virulence in China will be compared across a wider year span. The results should be useful in guiding resistance breeding programs and optimizing the strategic deployment of wheat cultivars carrying various genes for different types of resistance to stripe rust.

## 2. Materials and Methods

### 2.1. Collection of Wheat Stripe Rust Samples

During the annual survey of wheat stripe rust across China in 2020, leaf samples were collected from 13 provinces (Due to Beijing having only one sample and its proximity and highly similar agricultural practices to Hebei, we incorporated this sample into Hebei’s data for analysis in the following sections). The sampled infected leaves were air-dried at room temperature and kept in kraft paper bags stored at 4 °C for isolates purification and subsequent virulence tests. A total of 102 samples were selected from the collected samples to cover all regions and ensure at least two samples per area for subsequent purification and multiplication (Table 1 and Figure 1).

### 2.2. Purification and Multiplication of Isolates

Before purification, all samples were processed one more time to obtain adequate urediniospores that could be further used for purification. Specifically, leaf samples bearing uredinia were placed in a Petri dish containing distilled water for 24 h at 15 °C to promote sporulation. Then a 2 cm leaf segment bearing uredinia was cut from each sample and placed into a 2 mL centrifuge tube filled with Soltrol 170 isoparaffin (Chevron Phillips Chemical, Baytown, TX, USA) so that urediniospores from the leaf segment were suspended in the Soltrol solution. The urediniospore suspension with final concentration of 50 mg/mL was used to spray 7-day-old seedlings of the wheat cultivar Mingxian 169, which is susceptible to stripe rust, using a diaphragm vacuum pump (GM-0.5A, Jinteng Experimental Equipment, Tianjin, China). The inoculated seedlings were placed in a dark room with over 90% relative humidity at 9 °C for 24 h to facilitate infection. Following incubation, seedlings were transferred to a greenhouse with temperature kept at 15 °C during the 16 h light period and 10 °C during the 8 h dark period to allow for infection. To prevent cross-contamination, each pot was covered with a glossy paper bag (40 cm × 30 cm). After the emergence of uredinia, the step of purification was performed. For purification, a single uredinium from a previously inoculated leaf was isolated and inoculated again onto 7-day-old seedlings of the wheat variety ‘Mingxian 169’ using an inoculation needle. The inoculated seedlings were incubated as described above. Following incubation, seedlings were transferred to a greenhouse to allow for the multiplication of the pathogen. Wherever necessary, multiple rounds of multiplication were performed to obtain adequate urediniospores for virulence testing. Urediniospores from these subsequent inoculations were collected by tapping the paper bag, dried on a filter paper in a Petri dish, and stored at 4 °C until use.

### 2.3. Race Identification

Race identification was performed using the 18 *Yr* single-gene differentials as described by Wan and Chen [20]. Urediniospores were mixed with talcum powder at a 1:20 ratio and inoculated onto 8-day-old seedlings of the 18 *Yr* single-gene lines following the previously described procedures [20]. ‘Mingxian 169’ served as the susceptible check. After 18 days, when the susceptible check and differentials had fully sporulated, the infection types (ITs) on the first leaves were recorded. ITs were categorized as avirulent if they were 0 with no symptoms; 1 with necrotic or chlorotic flecks; 2 with necrotic or chlorotic spots or patches without sporulation; or 3–6 ranging from large necrotic or chlorotic patches and slight sporulation to moderate necrosis and moderate sporulation. In contrast, ITs of 7–9 with abundant sporulation ranging from slight necrosis or slight chlorosis to no necrosis/chlorosis were classified as virulent [20]. The races identified in this study were named as previously described in Zhang et al. [22].

### 2.4. Data Analysis

Virulence spectrum and frequency analysis were conducted using Virulence Analysis Tools (VAT) to identify the avirulence/virulence profiles of the isolates on the 18 *Yr* single-gene lines. The Nei genetic distances between races were calculated using VAT [23], and cluster analysis was conducted using the unweighted pair group method with arithmetic mean (UPGMA) in MEGA-10 software, as described in Zhang et al. [22]. All identified races were designated as PstCN and compared with the 2023 *Pst* populations in China to understand their genetic relationships and virulence variations [22].

## 3. Results

### 3.1. Virulence Patterns of Pst Races

In this study, 67 isolates were obtained from 102 collected samples (the remaining isolates were not viable during purification/multiplication), and a total of 33 races were identified (Table 2). The five predominant races were PstCN-062 (28.36%), PstCN-001 (11.94%), PstCN-002 (8.96%), PstCN-063 (5.97%), and PstCN-090 (2.99%). The remaining races each had a frequency below 2.99%. Interestingly, PstCN-002, PstCN-063, and PstCN-090 are variants of PstCN-062, which is equivalent to PSTv-137 in the U.S. *Pst* race. For example, PstCN-002 could be evolved from PstCN-062 by gaining additional virulence to *Yr27* and *Yr43*.

PstCN-062, with virulence to *Yr8*, *Yr44*, and *YrExp2*, had the broadest geographical distribution over 10 provinces. PstCN-077, PstCN-078, and PstCN-079 exhibited the broadest virulence spectra, each with ten virulence factors. Specifically, PstCN-077 was virulent to *Yr6*, *Yr7*, *Yr8*, *Yr9*, *Yr17*, *Yr27*, *Yr44*, *YrSP*, *YrExp2*, and *Yr76*; PstCN-078 to *Yr6*, *Yr7*, *Yr8*, *Yr9*, *Yr17*, *Yr27*, *Yr43*, *Yr44*, *YrSP*, and *YrExp2*; and PstCN-079 to *Yr6*, *Yr7*, *Yr8*, *Yr9*, *Yr27*, *Yr43*, *Yr44*, *YrSP*, *YrExp2*, and *Yr76*. In contrast, PstCN-085 (virulent to *Yr44* and *YrExp2*) and PstCN-090 (virulent to *Yr8* and *Yr44*) showed the fewest virulence factors. Notably, 28 other races were detected only once.

### 3.2. Frequencies of Virulence Factors

The *Pst* isolates from 2020 in China exhibited high virulence frequencies to *Yr8* (95.52%), *Yr44* (97.01%), and *YrExp2* (95.52%). Moderate virulence frequencies, ranging from 10% to 50%, were observed for *Yr6* (26.87%), *Yr7* (11.94%), *Yr9* (19.40%), *Yr17* (11.94%), *Yr27* (35.82%), and *Yr43* (38.81%). Low virulence frequencies, below 10%, were recorded for *Yr1* (4.48%), *Yr24* (2.99%), *YrSP* (8.96%), *Yr85* (1.49%), and *Yr76* (7.46%). Notably, no isolates exhibited virulence to *Yr5*, *Yr10*, *Yr15*, and *Yr32* (Figure 2).

We observed a significantly varied spectra of virulence among the *Pst* isolates collected from different geographical regions. For example, the isolates of Jiangsu and Chongqing provinces showed virulence to three *Yr* genes, while the isolates of Henan and Shanxi showed their virulence to ten *Yr* genes (Table 2). This variation in virulence shows the complex distribution of *Pst* and the importance of possible strategies to tailor the resistance in wheat cultivars.

### 3.3. Comparison of Pst Populations from China in 2020 and 2023

The 33 races identified from *Pst* isolates in 2020 in China had virulence spectra ranging from 2 to 10 virulence factors. In contrast, 61 races were identified in 2023, with a broader range of virulence spectra ranging from 3 to 16 virulence factors (Figure 3). Based on the virulence data, the virulence spectra of the races showed significant variation between the two years, with only five races being consistently detected in both years (Table 3).

Simultaneously, the frequency of virulence factors in the two years was compared. Among the 18 virulence loci, avirulence to *Yr5* and *Yr15* were not detected in both years. Notably, no virulence against *Yr10* was found in 2020. Only the virulence frequencies for *Yr8*, *Yr44*, and *YrExp2* showed minor differences, while the frequency differences for the other 12 virulence factors were obvious (Figure 4).

### 3.4. Correlations Between Virulence Factors

Analysis of virulence correlations among *Yr* genes revealed strong positive correlations (correlation coefficient > 0.7) between virulence factors to *Yr24* and *Yr85*, as well as between *YrSP* and *Yr76*. Conversely, moderate negative correlations (correlation coefficient < −0.1) were observed between virulence factors to *Yr7* and *Yr44*, *Yr8* and *Yr17*, *Yr43* and *Yr85*, and *Yr44* and *Yr76* (Table 4). These correlations suggest potential interactions and associations among virulence factors to the resistance genes, which could be used in breeding strategies with appropriate combinations of resistance genes.

### 3.5. Pst Race Clustering

Based on the virulence patterns observed in the 18 *Yr* single-gene lines, the 33 identified *Pst* races were clustered into seven distinct virulence groups (VGs) using Nei’s genetic distance (Figure 5). VG1 comprising seven races, characterized by virulence to *Yr8*, *Yr27*, *Yr43*, *Yr44*, and *YrExp2*, with *Yr8*, *Yr44*, and *YrExp2* being the predominant virulence factors in this group. VG2 included three races with virulence to *Yr6*, *Yr8*, *Yr9*, *Yr44* and *YrExp2*, with virulence to *Yr8* as the dominant factor. VG3 consisted of two races showing virulence to *Yr7*, *Yr8*, *Yr44* and *YrExp2*, with *Yr7*, *Yr8*, and *YrExp2* being the predominant virulence factors. VG4 consisted of three races showing virulence to *Yr1*, *Yr6*, *Yr7*, *Yr8*, *Yr9*, *Yr27*, *Yr44*, and *YrExp2*, with virulence to *Yr6*, *Yr27*, *Yr44*, and *YrExp2* as common factors. VG5 contained six races virulent to *Yr1*, *Yr6*, *Yr8*, *Yr9*, *Yr24*, *Yr27*, *Yr43*, *Yr44*, and *YrExp2*, with the *Yr43* virulence factor being predominant. VG6 had a single race, virulent to *Yr8*, *Yr27*, *Yr43*, *YrExp2*, and *Yr76*. VG7 comprised three races showing virulence to *Yr6*, *Yr8*, *Yr9*, *Yr17*, *Yr24*, *Yr27*, *Yr44*, *YrExp2*, and *Yr85*, with virulence to *Yr9*, *Yr44*, and *YrExp2* being the predominant factors. VG8 consisted of eight races virulent to *Yr6*, *Yr7*, *Yr8*, *Yr9*, *Yr17*, *Yr27*, *Yr43*, *Yr44*, *YrSP*, *YrExp2*, and *Yr76*, of which virulence factors to *YrSP* and *Yr76* were predominant.

## 4. Discussion

### 4.1. Races and Epidemics

Our study identified PstCN-062 as the most predominant race, with a widespread distribution across 10 provinces. These regions have predominantly grown wheat cultivars Jimai, Zhengmai, Huamai, and Qinmai [24,25], with resistance genes *Yr9*, *Yr10*, *Yr15*, and *Yr32*. Notably, PstCN-062 (virulent to *Yr8*, *Yr44* and *YrExp2*) lacks virulence to these resistance genes, reducing its potential to cause widespread susceptibility in these wheat cultivars.

The predominant races identified in China using traditional cultivar differentials are CYR32 and CYR34 [6]. To explore the relationships between PstCN-062 and these established races, we conducted virulence tests on Chinese differentials with a subset of six isolates from the PstCN-062 group. Among these, one isolate from Gansu province exhibited a virulence profile similar to CYR32, which is virulent on Hybrid 46, while the remaining isolates were avirulent to Hybrid 46. This suggests that while PstCN-062 shares some virulence characteristics with CYR32 and potentially CYR34, it does not align precisely with these CYR races.

While the presence of common PstCN races among different provinces suggests the transmission of *Pst* among regions in China, the province-specific races may indicate the local adaptation of *Pst* to agricultural practices (i.e., utilization of specific resistance genes). A spatiotemporal evaluation of *Pst* virulence across years and geographical regions will be needed to assess the selection pressure imposed by resistance genes currently deployed in regional agricultural systems on *Pst* population.

Our results exhibited a high virulence diversity of *Pst* in China in 2020, which warrants continuous monitoring of new races evolved through sexual reproduction on its alternate hosts, *Berberis* spp., somatic recombination or mutations. This information may help to understand the mechanism of changes in virulence and future breeding strategies for new resistant wheat cultivars.

### 4.2. Effectiveness of Resistance Genes and Gene Interactions

We evaluated the effectiveness of 18 *Yr* resistance genes against *Pst* races and found no isolates virulent to *Yr5*, *Yr10*, *Yr15*, or *Yr32*. This finding is consistent with the previous study by Li et al. in 2016 [21]. Previous studies up to 2016 reported no commercial varieties carrying these genes [26,27,28,29], but a 2020 report indicated that multiple wheat lines have these genes [30], indicating that these genes have been recently used in breeding programs. Combining them with other effective resistance genes into elite breeding lines could develop wheat cultivars with durable resistance to stripe rust and desirable agronomic and quality traits. *Yr85* (*YrTr1*) and *Yr24* also remain effective, with low virulence frequencies of 1.49% and 2.99%, respectively. Despite its effectiveness, vigilance is necessary to monitor potential new races emerging from host–pathogen interactions.

In contrast, virulence factors to *Yr8*, *Yr44*, and *YrExp2* demonstrated high frequencies, exceeding 90%, showing that these genes are not effective against the Chinese *Pst* population. *Yr8* was initially introduced into common wheat through hybridization between *Aegilops comosa* and ‘Chinese Spring’ in 1968 [31]. Virulence to *Yr8* was first identified in the United States in 1998 at a frequency of 7.5%, which reached 90.9% by 2009 [32]. In the present study, we found that the *Yr8* virulence was also very high, at 95.6%. Li et al. reported that the race-specific all-stage resistance gene *Yr8* is linked to a non-race specific high-temperature adult-plant resistance gene from *Ae. comosa*, making the source of different types of resistance still useful in breeding stripe rust resistant wheat cultivars [21]. *YrExp2*, identified via QTL mapping in 2009 from a cross between ‘Express’ and ‘Avocet S’, has been extensively used in the United States. The *YrExp2* virulence existed in the 1960s, although at a low frequency of 3.8% in 1968, and it reached a high frequency of 88.6% by 2009 in the United States [32]. *Yr44*, identified from the U.S. spring wheat cultivar ‘Zak’ [33]. The *Yr44* virulence was detected from the 1968 *Pst* collection and reached to 90.9% in 2009 in the United States [32]. In the present study, 97.01% of the isolates were found virulent to *Yr44* indicating its ineffectiveness in China. These genes should not be intentionally used in breeding programs.

The correlations observed between different resistance genes suggest strategic opportunities for gene stacking. Genes with negative virulence correlations, such as *Yr7* and *Yr44*, *Yr8* and *Yr17*, *Yr43* and *Yr85*, *Yr44* and *Yr76*, could be used in combination, but caution is still advised. In contrast, resistance genes with their virulence factors positively associated, such as genes *Yr24* and *Yr85*, and *YrSP* and *Yr76*, should not be used in combinations in breeding programs may increase the breakdown risk of resistance. More importantly, the predominant races and those representing different race groups should be used in screening wheat germplasm, especially those carrying reported stripe rust resistance genes, to select germplasm or individual genes for effective resistance to be used in breeding programs.

### 4.3. Virulence Dynamics of Pst Populations

The virulence frequencies of Chinese (CN) *Pst* isolates to the 18 *Yr* genes between 2020 and 2023 are shown in Figure 4. A comparative analysis revealed marked differences in virulence profiles of the *Pst* populations, with virulence frequency differences exceeding 50% for resistance genes *Yr1*, *Yr6*, *Yr9*, *Yr17*, and *Yr43*. Substantial fluctuations were observed in the prevalence of virulence factors. Of epidemiological significance is the consistent absence of *Yr5*-virulent and *Yr15*-virulent races across both sampling years. Future work on continuously monitoring of virulence of *Pst* populations in China will provide more details on effectiveness of *Yr* genes.

Comparative virulence profiling revealed that *Pst* isolates from the United States exhibited significantly elevated virulence prevalence against *Yr1*, *Yr6*, *Yr7*, *Yr9*, *Yr10*, *Yr17*, *Yr27*, *Yr32*, *Yr43*, and *Yr85* compared to the Chinese *Pst* populations Appendix A. Strikingly, both *Pst* populations maintained almost complete avirulence to *Yr5* and *Yr15*, confirming the broad effectiveness of this gene across geographical and temporal scales. The *Yr5* virulence has been reported recently in China [33,34] and Turkey [35], but still at very low frequencies. Although *Yr5* is largely effective, the gene should be used in combination with other effective genes for long-lasting resistance.

The 18 *Yr* single-gene differentials have been employed in the United States for years. The present study used the same differentials to assess the virulence of the *Pst* population in China in 2020, allowing for a comparison for the pathogen virulence between the two countries. Our results indicate significant differences between the Chinese and U.S. *Pst* populations. In 2020, the predominant race in the United States was PSTv-37 with virulence to *Yr6*, *Yr7*, *Yr8*, *Yr9*, *Yr17*, *Yr27*, *Yr43*, *Yr44*, *Yr85*, and *YrExp2* and avirulence to *Yr1*, *Yr5*, *Yr10*, *Yr15*, *Yr24*, *Yr32*, *YrSP*, and *Yr76*, accounting for 54.2% of the U.S. isolates in 2020 [36]. This race has not been detected in China. The Chinese races identified in 2020 with the closest virulence spectra to PSTv-37 were PstCN-078 and PstCN-081, with the former being virulent to *YrSP* and both avirulent to *Yr85*. The virulence frequency of these races in China was only 1.49%.

Liu et al. tested the U.S. *Pst* isolates collected from 1968 to 2009 using the 18 *Yr* single-gene differentials and reported a much broader range of virulence spectra than the range in the present study in China in 2020 [32]. The difference could be attributed to several factors. First, in their study, more than 800 isolates collected from 42 years were tested, while the present study used only 67 isolates from only one year, which limits direct comparability between the two studies, the *Pst* pathogens in the U.S. and China may have distinct genetic backgrounds, as significant genetic divergence and differences in virulence and molecular structures in the previous studies [32]. Third, differences in wheat cultivars and environmental factors could contribute to different pathogen-host interactions [36,37,38,39]. Understanding these differences in virulence composition can facilitate the introduction of superior foreign wheat germplasm for breeding wheat cultivars with stripe rust resistance.

## 5. Conclusions

This study utilized the 18 *Yr* single-gene lines established by [36] to characterize virulence in 67 *Pst* isolates collected from 13 provinces in China in 2020. A total of 33 races were identified. Notably, none of the isolates were virulent to *Yr5*, *Yr10*, *Yr15*, and *Yr32*, underscoring the continued effectiveness of these resistance genes. Stacking or pyramiding of these low-virulence genes has great potential for enhancing resistance durability to *Pst*, especially coupling with cultural practices (e.g., eradication of alternate hosts of *Pst* wherever possible). Frequencies of virulence to *Yr1* (4.48%), *Yr24* (2.99%), *YrSP* (8.96%), *Yr85* (1.49%), and *Yr76* (7.46%) remained below 10%; moderate frequencies were observed for virulence to *Yr6* (26.87%), *Yr7* (11.94%), *Yr9* (19.40%), *Yr17* (11.94%), *Yr27* (35.82%), and *Yr43* (38.81%); and high virulence frequencies were recorded for *Yr8*, *Yr44*, and *YrExp2*, all exceeding 90%. The resistance genes with moderate and high virulence frequencies should be used cautiously or avoided in future breeding efforts. Among the 33 races, the most prevalent races were PstCN-062, PstCN-001, PstCN-002, PstCN-063, and PstCN-090. The widespread distribution of these races, particularly PstCN-062, highlights the virulence nature of the *Pst* population and the need for ongoing surveillance. Comparative analysis with the 2023 *Pst* population’s virulence revealed two key findings: (1) five races consistently detected across both years; and (2) significant shifts in virulence frequency for 12 *Yr* genes. Continuous monitoring and surveillance should help combat the threat of new virulent races, safeguard wheat production, and contribute to global food security.

## Figures and Tables

**Figure 1 jof-11-00447-f001:**
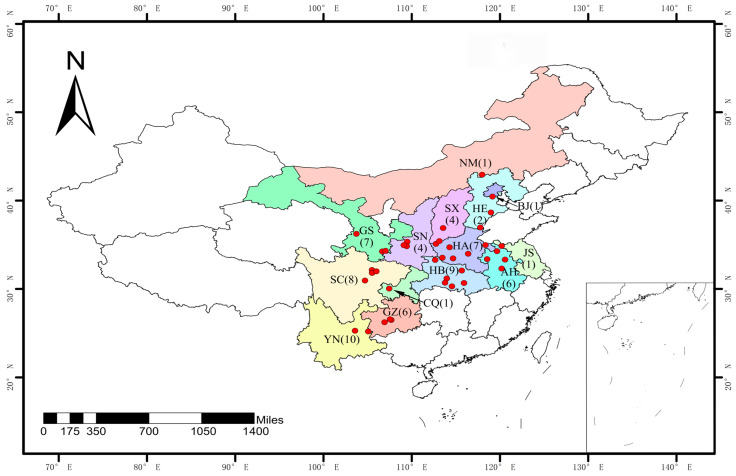
Map showing the locations and numbers of stripe rust (caused by *Puccinia striiformis* f. sp. *tritici*) samples collected from 13 provinces in 2020 in China. The number of isolates in each province is shown in parenthesis and the red dots indicate the locations. The 13 provinces are HE: Hebei, HA: Henan, SX: Shanxi, SN: Shaanxi, SC: Sichuan, GS: Gansu, AH: Anhui, JS: Jiangsu, GZ: Guizhou, HB: Hubei, YN: Yunnan, CQ: Chongqing, and NM: Inner Mongolia.

**Figure 2 jof-11-00447-f002:**
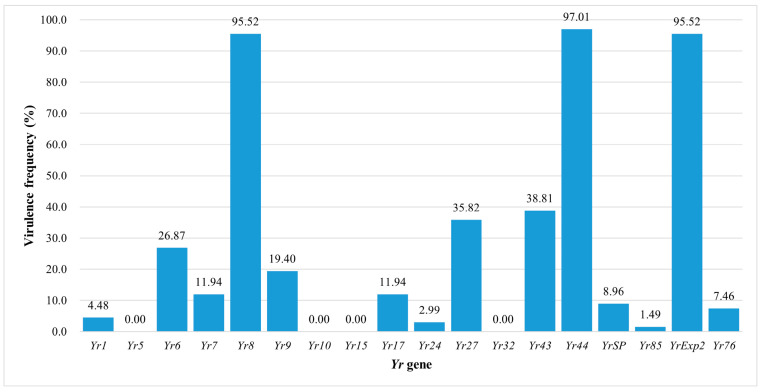
Frequencies of virulence factors of *Puccinia striiformis* f. sp. *tritici* isolates to the 18 *Yr* single-gene differentials in China in 2020.

**Figure 3 jof-11-00447-f003:**
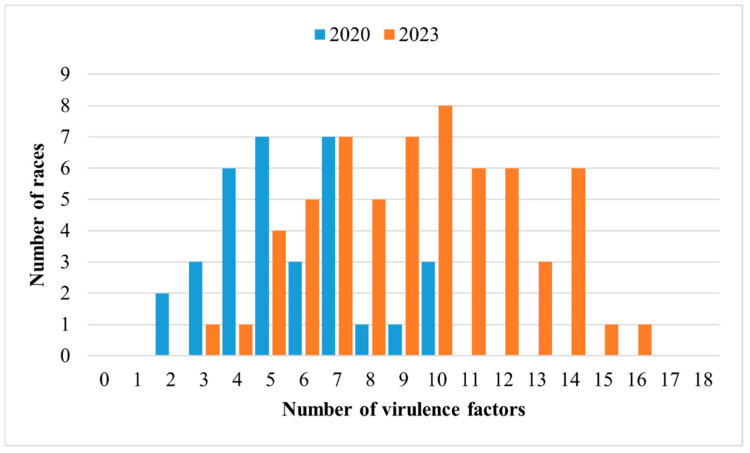
Distributions of *Puccinia striiformis* f. sp. *tritici* races with different numbers of virulence factors in 2020 and 2023.

**Figure 4 jof-11-00447-f004:**
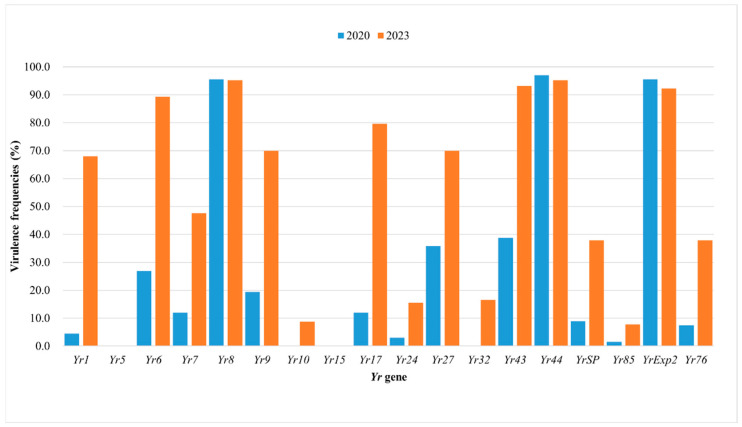
The virulence frequencies of *Puccinia striiformis* f. sp. *tritici* (*Pst*) in China in 2020 and 2023.

**Figure 5 jof-11-00447-f005:**
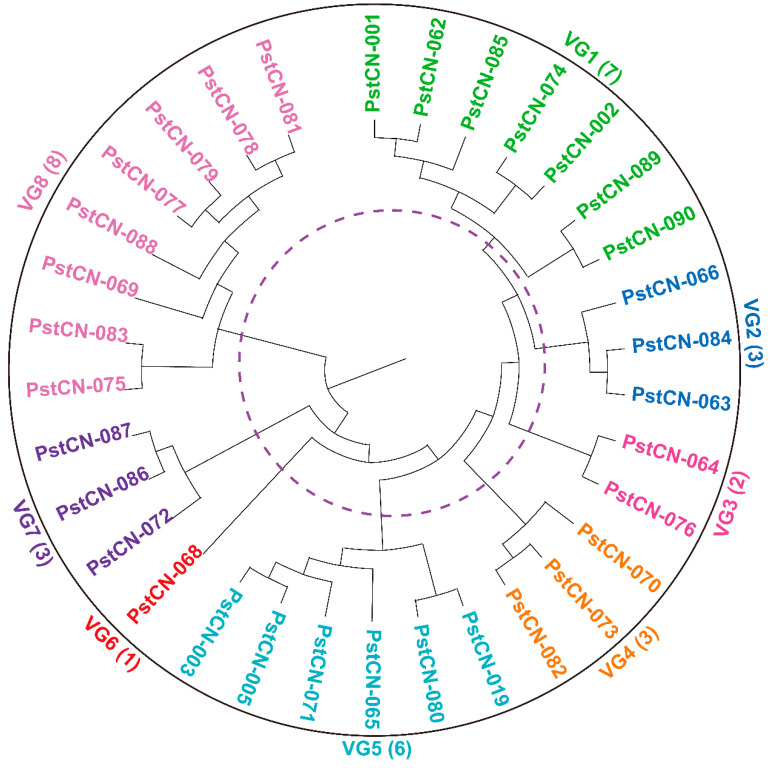
Virulence relationships of 33 *Puccinia striiformis* f. sp. *tritici* races identified from the 2020 collection in China based on the avirulent/virulent data on the 18 *Yr* single-gene differentials using the unweighted pair group method with arithmetic means and Nei’s genetic distance. VG = virulence group.

**Table 1 jof-11-00447-t001:** The number of samples received, isolates, and races of *Puccinia striiformis* f. sp. *tritici* identified from China in 2020.

Province ^1^	No. of Cities	No. of Samples	No. of Isolates	Races	Isolate/Race Ratio
HE	7	8	3	3	1.00
HA	7	13	7	5	1.40
SX	3	6	4	4	1.00
SN	3	7	4	4	1.00
SC	3	11	8	6	1.33
GS	2	11	7	5	1.40
AH	4	6	6	4	1.50
JS	5	2	1	1	1.00
GZ	2	8	6	5	1.20
HB	9	13	9	8	1.12
YN	3	13	10	8	1.25
CQ	1	2	1	1	1.00
NM	1	2	1	1	1.00

^1^ HE: Hebei, HA: Henan, SX: Shanxi, SN: Shaanxi, SC: Sichuan, GS: Gansu, AH: Anhui, JS: Jiangsu, GZ: Guizhou, HB: Hubei, YN: Yunnan, CQ: Chongqing, NM: Inner Mongolia.

**Table 2 jof-11-00447-t002:** Races of *Puccinia striiformis* f. sp. *tritici* and their octal codes, virulence patterns, numbers, frequencies, and distributions in China in 2020.

Race	PSTv ^1^	Octal Caode	Virulence to *Yr* Gene	No. of Virulence Factors	No. of Isolates	Frequency (%) (*n* = 67)	Province ^2^ (Number of Isolates)
PstCN-001	PSTv-136	020062	8, 43, 44, Exp2	4	8	11.94	YN(2), SC(3), SN(2), HE(1)
PstCN-002	\	020262	8, 27, 43, 44, Exp2	5	6	8.98	HB(1), GS(2), HA(1), YN(1), SC(1)
PstCN-003	\	120062	6, 8, 43, 44, Exp2	5	1	1.49	AH(1)
PstCN-005	\	120262	6, 8, 27, 43, 44, Exp2	6	1	1.49	SC(1)
PstCN-019	\	420062	1, 8, 43, 44, Exp2	5	1	1.49	GZ(1)
PstCN-062	PSTv-137	020022	8, 44, Exp2	3	19	28.36	AH(3), GS(2), GZ(2), HA(3), HB(2), JS(1), SX(1), SN(1), SC(2), YN(2)
PstCN-063	\	120022	6, 8, 44, Exp2	4	4	5.97	GS(1), SN(1), HE(1), HB(1)
PstCN-064	\	060002	7, 8, Exp2	3	1	1.49	HB(1)
PstCN-065	\	100062	6, 43, 44, Exp2	4	1	1.49	YN(1)
PstCN-066	\	030022	8, 9, 44, Exp2	4	1	1.49	HB(1)
PstCN-068	\	020243	8, 27, 43, Exp2, 76	5	1	1.49	HA(1)
PstCN-069	\	061262	7, 8, 17, 27, 43, 44, Exp2	7	1	1.49	HB(1)
PstCN-070	\	530222	1, 6, 8, 9, 27, 44, Exp2	7	1	1.49	GZ(1)
PstCN-071	\	120662	6, 8, 24, 27, 43, 44, Exp2	7	1	1.49	YN(1)
PstCN-072	\	030626	8, 9, 24, 27, 44, 85, Exp2	7	1	1.49	SX(1)
PstCN-073	\	120222	6, 8, 27, 44, Exp2	5	1	1.49	GZ(1)
PstCN-074	\	020222	8, 27, 44, Exp2	4	1	1.49	SC(1)
PstCN-075	\	031233	8, 9, 17, 27, 44, SP, Exp2, 76	8	1	1.49	HB(1)
PstCN-076	\	060022	7, 8, 44, Exp2	4	1	1.49	YN(1)
PstCN-077	\	171233	6, 7, 8, 9, 17, 27, 44, SP, Exp2, 76	10	1	1.49	HA(1)
PstCN-078	PSTv-323	171272	6, 7, 8, 9, 17, 27, 43, 44, SP, Exp2	10	1	1.49	SX(1)
PstCN-079	\	170273	6, 7, 8, 9, 27, 43, 44, SP, Exp2, 76	10	1	1.49	NM(1)
PstCN-080	\	530062	1, 6, 8, 9, 43, 44, Exp2	7	1	1.49	HA(1)
PstCN-081	PSTv-052	171262	6, 7, 8, 9, 17, 27, 43, 44, Exp2	9	1	1.49	HB(1)
PstCN-082	\	160222	6, 7, 8, 27, 44, Exp2	6	1	1.49	YN(1)
PstCN-083	\	021033	8, 17, 44, SP, Exp2, 76	6	1	1.49	AH(1)
PstCN-084	\	130022	6, 8, 9, 44, Exp2	5	1	1.49	AH(1)
PstCN-085	PSTv-130	000022	44, Exp2	2	1	1.49	YN(1)
PstCN-086	\	131222	6, 8, 9, 17, 27, 44, Exp2	7	1	1.49	GS(1)
PstCN-087	\	011222	9, 17, 27, 44, Exp2	5	1	1.49	SX(1)
PstCN-088	\	030272	8, 9, 27, 43, 44, SP, Exp2	7	1	1.49	GS(1)
PstCN-089	\	020220	8, 27, 44	3	1	1.49	HE(1)
PstCN-090	\	020020	8, 44	2	2	2.99	GZ(1), CQ(1)

^1^ ‘\’ denotes that the race has not been found in the United States and other countries (https://striperust.wsu.edu (accessed on 20 September 2022)). ^2^ HE: Hebei, HA: Henan, SX: Shanxi, SN: Shaanxi, SC: Sichuan, GS: Gansu, AH: Anhui, JS: Jiangsu, GZ: Guizhou, HB: Hubei, YN: Yunnan, CQ: Chongqing, NM: Inner Mongolia.

**Table 3 jof-11-00447-t003:** The number of isolates (No.), virulence frequencies (Fre.) and distribution of 5 races in China in 2020 and 2023.

Race	Octal	Virulence to *Yr* Gene	2020 (N = 67)			2023 (N = 103)		
			No.	Fre. (%)	Distribution	No.	Fre. (%)	Distribution
PstCN-001	020062	8, 43, 44, Exp2	8	11.94	YN(2), SC(3), SN(2), HE(1)	5	4.85	SC(4), GZ(1)
PstCN-002	020262	8, 27, 43, 44, Exp2	6	8.96	HB, GS(2), HA, YN, SC	1	0.97	QH(1)
PstCN-003	120062	6, 8, Exp2, 44, 43	1	1.49	AH	3	2.91	HB(1), GS(1), AH(1)
PstCN-005	120262	8, 6, 27, Exp2, 44, 43	1	1.49	SC	3	2.91	AH(2), SC(1)
PstCN-019	420062	1, 8, 43, Exp2, 44	1	1.49	GZ	1	0.97	HE(1)

HE = Hebei, HA = Henan, SX = Shanxi, SN = Shaanxi, SC = Sichuan, GS = Gansu, AH = Anhui, JS = Jiangsu, GZ = Guizhou, HB = Hubei, YN = Yunnan, CQ = Chongqing, NM = Inner Mongolia.

**Table 4 jof-11-00447-t004:** The correlation coefficients of virulence to 18 *Yr* genes determined by *Puccinia striiformis* f. sp. *tritici* isolates in China in 2020.

	Virulence to *Yr* Gene																
Vir to	*Yr5*	*Yr6*	*Yr7*	*Yr8*	*Yr9*	*Yr10*	*Yr15*	*Yr17*	*Yr24*	*Yr27*	*Yr32*	*Yr43*	*Yr44*	*YrSP*	*Yr85*	*YrExp2*	*Yr76*
*Yr1*	0.00	0.21	−0.08	0.05	0.26	0.00	−0.03	−0.08	−0.04	−0.02	0.00	0.13	0.04	−0.07	−0.03	0.05	−0.06
*Yr5*		0.00	0.00	0.00	0.00	0.00	0.00	0.00	0.00	0.00	0.00	0.00	0.00	0.00	0.00	0.00	0.00
*Yr6*			0.32	−0.04	0.32	0.00	−0.07	0.21	0.10	0.26	0.00	0.11	0.10	0.18	−0.07	0.12	0.10
*Yr7*				0.08	0.29	0.00	−0.05	0.43	−0.06	0.29	0.00	0.09	−0.21	0.37	−0.05	0.08	0.25
*Yr8*					−0.08	0.00	0.03	−0.14	0.04	0.02	0.00	0.02	−0.04	0.07	0.03	−0.05	0.06
*Yr9*						0.00	−0.06	0.52	0.14	0.41	0.00	0.00	0.09	0.51	0.25	0.10	0.29
*Yr10*							0.00	0.00	0.00	0.00	0.00	0.00	0.00	0.00	0.00	0.00	0.00
*Yr15*								0.00	0.00	0.00	0.00	0.00	0.00	0.00	0.00	0.00	0.00
*Yr17*									−0.06	0.38	0.00	−0.01	0.06	0.53	−0.05	0.08	0.42
*Yr24*										0.23	0.00	0.04	0.03	−0.05	0.70	0.04	−0.05
*Yr27*											0.00	0.28	−0.05	0.30	0.16	0.02	0.25
*Yr32*												0.00	0.00	0.00	0.00	0.00	0.00
*Yr43*													−0.04	0.08	−0.10	0.17	0.01
*Yr44*														0.05	0.02	−0.04	−0.28
*YrSP*															−0.04	0.07	0.71
*Yr85*																0.03	−0.03
*YrExp2*																	0.06

## Data Availability

The original contributions presented in this study are included in the article/Appendix A. Further inquiries can be directed to the corresponding authors.

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
