# Peer review of "Virulence Characterization of Puccinia striiformis f. sp. tritici in China in 2020 Using Wheat Yr Single-Gene Lines"

_jof, 2025, doi:10.3390/jof11060447_

Round 1
Reviewer 1 Report
The article presents studies on the virulence of Puccinia striiformis (Pst) populations from different provinces of China. The results obtained are of great importance for the management of wheat rust, for the development of resistant varieties to different races. The authors have presented a good overview of the dynamics of races and virulence of populations of the yellow rust pathogen from the 1950s to the present. The methods are described in detail and clearly, the results correspond to the purpose of the study, the conclusions are justified and specific. There is a general wish and a comment on the presentation of data for 2023.
The title of the article and the main results present the results of the study of the virulence of Pst isolates for 2020. However, the results present data that compare and describe the virulence and racial composition for 2020 and 2023. This is also very interesting data that can and should be shown, but then it needs to be justified in the introduction, materials and methods, and possibly in the title. Otherwise, it is not clear why exactly 2023, and not 2024 or 2022? After this comment is corrected, the article can be accepted for publication.
Reviewer 2 Report
There is a great diversity of virulence genes of the pathogen in China, probably due to the existence of the sexual phase of P. striiformis f. sp. tritici. Given this situation and considering that the resistance of wheat cultivars has not been durable, the authors could improve the discussion on how to make resistance to this disease more durable. Whether by improving the search for horizontal resistance, increasing the diversity of resistance genes, or cultural practices that can help control the disease.
Specific comments
Line 33: Consistency of keywords and title. The term "Puccinia striiformis f. sp. tritici" appears in both the keywords and the title. Consider rewording the keywords to avoid redundancy. In addition, I suggest that the scientific name of wheat be used.
Line 176: The graphic presentation of this figure needs to be improved. Specifically, the resolution needs to be improved. Provide a clear description of what each column represents in the figure legend. In addition, error bars should be added to the data points, and the color scheme should be standardized across all figures in the article to ensure visual consistency.
Line 302: Figure 4 is in the Discussion section. Reconsider organizing it into the Results section.
Row 337: The conclusion section could be strengthened by exploring the data collected in 2023. Discuss the importance of comparing this data with the 2020 results, which is highlighted in the title of the article. A concise summary of this comparison and its implications should be integrated into the conclusion as well as the abstract to provide a more comprehensive overview of the research.

Reviewer 3 Report
This manuscript presents a scientifically sound and well-organized study on the virulence profiling of Puccinia striiformis f. sp. tritici (Pst) in China using 18 Yr single-gene differentials. Based on 67 isolates from 13 provinces, the authors identified 33 races, offering important insights into pathogen diversity and resistance gene effectiveness. The work is timely and valuable for breeding, surveillance, and global comparative studies, but still requires some minor clarifications and language revisions.
Revisions and Suggestions:
- Were the PstCN race names (e.g., PstCN-001, PstCN-062) newly assigned in this study? Please provide a short description of the naming convention used and whether it builds on or diverges from the historical CYR system.
- The abstract and methods state 13 provinces, but the conclusion mentions 14. Please verify and ensure consistency throughout the manuscript.
- Clarify the impact of combining the single Beijing sample with Hebei in both the methodology and any regional analysis.
Improve phrasing for better academic tone. Examples:
- “Resistance deteriorated from immunity to susceptibility” → “Resistance transitioned from full immunity to susceptibility.”
- “...but still need to be cautions” → “...but caution is still advised.”
- “...which makes the two studies less comparable” → “...which limits direct comparability between the two studies.”
- In the conclusion, rephrase “should not be intentionally used” to “should be used cautiously or avoided in future breeding efforts.”
- Consider mentioning the potential for gene stacking/pyramiding of low-virulence genes (Yr5, Yr10, Yr15, Yr32) to enhance durability.
- For newly noted races such as PstCN-002, PstCN-063, and PstCN-090, briefly indicate how their virulence profiles compare to PstCN-062. Are they variants or represent novel combinations?
Questions for Authors:
- How were the 67 isolates selected from the original 102 samples?
- Was the selection random, or based on geographical and host cultivar diversity?
- What was the rationale for using the 18 Yr single-gene lines?
- Are these lines part of Chinese breeding pipelines or used for international standardization?
- How do you plan to manage future naming and monitoring of PstCN races?
- Is there a national registry or systematic protocol for aligning new races with historical CYR designations?
- Have any of the PstCN races shown continuity in surveys from 2021 to 2023?
- If so, which races persist across multiple years, and what does this imply for virulence stability?
- Are any predominant races capable of overcoming adult plant resistance (APR) genes, such as Yr18 or high-temperature adult plant resistance?
This manuscript presents a scientifically sound and well-organized study on the virulence profiling of Puccinia striiformis f. sp. tritici (Pst) in China using 18 Yr single-gene differentials. Based on 67 isolates from 13 provinces, the authors identified 33 races, offering important insights into pathogen diversity and resistance gene effectiveness. The work is timely and valuable for breeding, surveillance, and global comparative studies, but still requires some minor clarifications and language revisions.
Revisions and Suggestions:
- Were the PstCN race names (e.g., PstCN-001, PstCN-062) newly assigned in this study? Please provide a short description of the naming convention used and whether it builds on or diverges from the historical CYR system.
- The abstract and methods state 13 provinces, but the conclusion mentions 14. Please verify and ensure consistency throughout the manuscript.
- Clarify the impact of combining the single Beijing sample with Hebei in both the methodology and any regional analysis.
Improve phrasing for better academic tone. Examples:
- “Resistance deteriorated from immunity to susceptibility” → “Resistance transitioned from full immunity to susceptibility.”
- “...but still need to be cautions” → “...but caution is still advised.”
- “...which makes the two studies less comparable” → “...which limits direct comparability between the two studies.”
- In the conclusion, rephrase “should not be intentionally used” to “should be used cautiously or avoided in future breeding efforts.”
- Consider mentioning the potential for gene stacking/pyramiding of low-virulence genes (Yr5, Yr10, Yr15, Yr32) to enhance durability.
- For newly noted races such as PstCN-002, PstCN-063, and PstCN-090, briefly indicate how their virulence profiles compare to PstCN-062. Are they variants or represent novel combinations?
Questions for Authors:
- How were the 67 isolates selected from the original 102 samples?
- Was the selection random, or based on geographical and host cultivar diversity?
- What was the rationale for using the 18 Yr single-gene lines?
- Are these lines part of Chinese breeding pipelines or used for international standardization?
- How do you plan to manage future naming and monitoring of PstCN races?
- Is there a national registry or systematic protocol for aligning new races with historical CYR designations?
- Have any of the PstCN races shown continuity in surveys from 2021 to 2023?
- If so, which races persist across multiple years, and what does this imply for virulence stability?
- Are any predominant races capable of overcoming adult plant resistance (APR) genes, such as Yr18 or high-temperature adult plant resistance?
Reviewer 4 Report
The manuscript communicates relevant research work, including the virulence characterization of Puccinia striiformis f. sp. tritici in samples collected in 2020 in different regions of China. The survey of races and virulence profiles in isolates of this fungal species generates valuable information when selecting wheat strains to be used in the field. The authors found changes in a survey from 2023, highlighting the need for this kind of study.
My only major criticism is related to strain preservation after being collected from the fields. The authors need to provide details in this regard. This subject is relevant discard changes in virulence associated with the wrong techniques of strain preservation.
As minor points, the methodology is too succinct and does not allow reproducibility of the results. The authors are invited to expand and provide more technical details. In addition, the discussion should be expanded and include the relevance of the findings in the different Chinese regions, and what is the consequence of finding certain races in particular locations?
The manuscript communicates relevant research work, including the virulence characterization of Puccinia striiformis f. sp. tritici in samples collected in 2020 in different regions of China. The survey of races and virulence profiles in isolates of this fungal species generates valuable information when selecting wheat strains to be used in the field. The authors found changes in a survey from 2023, highlighting the need for this kind of study.
My only major criticism is related to strain preservation after being collected from the fields. The authors need to provide details in this regard. This subject is relevant discard changes in virulence associated with the wrong techniques of strain preservation.
As minor points, the methodology is too succinct and does not allow reproducibility of the results. The authors are invited to expand and provide more technical details. In addition, the discussion should be expanded and include the relevance of the findings in the different Chinese regions, and what is the consequence of finding certain races in particular locations?
Round 2
Reviewer 2 Report
The work is well written, the methodology and results are consistent and can contribute to improving genetic improvement programs in obtaining cultivars with more durable resistance to rust.
The previously suggested corrections have been addressed
Author Response
Comment 1: The work is well written, the methodology and results are consistent and can contribute to improving genetic improvement programs in obtaining cultivars with more durable resistance to rust.
Response: Thanks.
Reviewer 4 Report
I thank the authors for the manuscript improvement. However, the methodology still has room for improvement. Please include details to allow experimental reproducibility.
I thank the authors for the manuscript improvement. However, the methodology still has room for improvement. Please include details to allow experimental reproducibility.
Author Response
Comment 1: I thank the authors for the manuscript improvement. However, the methodology still has room for improvement. Please include details to allow experimental reproducibility.
Response: Thanks again for the suggestion. In this version, we re-write the '2.2 Purification and multiplication of isolates' section, we added all steps that are necessary to reproduce this experiment. Please check Lines 121-145.